# Clinical and laboratory characteristics of adolescents and young adults with sickle cell disease at steady state in Uganda

Abdul Magala Ssekandi[1,11]*, Ruth Namazzi[3], Haruna Muwonge[1,2,11], Robert Kalyesubula[1], Ian Guyton Munabi[1,4], Christine Sekaggya-Wiltshere[5,6], Agnes Namaganda[1,11], Ivan Kimuli[1,2], Roselyne Akugizibwe[2,5,11], Hassan Kasujja[8,9,10,11], David Mukunya[3,7], Grace Ndeezi[3], Sarah Kiguli[3]

**1** Department of Physiology, School of Biomedical Sciences, College of Health Sciences, Makerere University, Kampala, Uganda, **2** Makerere Lung Institute, Kampala, Uganda **3** Department of Pediatrics and Child Health, School of Medicine, College of Health Sciences, Makerere University, Kampala, Uganda, **4** Department of Anatomy, School of Biomedical Sciences, College of Health Sciences, Makerere University, Kampala, Uganda, **5** Department of Internal Medicine, School of Medicine, College of Health Sciences, Makerere University, Kampala, Uganda, **6** Infectious Diseases Institute, Makerere University, Kampala, Uganda **7** Department of Community and Public Health, Faculty of Health Sciences, Mbale, Uganda, **8** Clarke International University, Kampala, Uganda **9** Habib Medical School, Islamic University in Uganda, Mbale, Uganda, **10** Department of Immunology and Molecular Biology, School of Biomedical Sciences, College of Health Sciences, Makerere University, Kampala, Uganda, **11** Translational Research Initiative for Biomedical Science Excellence (TRIBE) Research group, College of Health Sciences, Makerere University, Kampala, Uganda

* ssekandi91@gmail.com

## Abstract

Sickle cell disease (SCD) is associated with chronic systemic morbidity that extends beyond acute crises. However, data describing the clinical and laboratory characteristics of adolescents and young adults with SCD at steady state in sub-Saharan Africa are limited. We described clinical and laboratory characteristics of adolescents and young adults with SCD at steady state in Uganda. We conducted a hospital-based cross-sectional study of 60 adolescents and young adults with SCD in steady state at Mulago National Referral Hospital. Descriptive statistics were used to summarize participant characteristics and medication use. The mean age was 16.5±3.3 years, and 34 (56.7%) participants were female. Mean hemoglobin was 9.1±2.2 g/dl. Mean systolic and diastolic blood pressures were 107.9±15.5 mmHg and 60.3±12.6 mmHg, respectively; mean heart rate was 89.5±15.5 beats/min. Fifty-two (86.7%) participants reported using hydroxyurea. These observations show that adolescents and young adults with SCD at steady state exhibit hematologic abnormalities and distinctive hemodynamic profiles that underscore substantial chronic subclinical abnormalities that extend beyond acute complications.

**Data availability statement:** Dataset is available at https://doi.org/10.17605/OSF.IO/P837G.

**Funding:** This work was was supported by the Fogarty International Center of the National Institutes of Health under Award Number 1R25TW011213 (to SK). The content is solely the authors' responsibility and does not necessarily represent the official views of the National Institutes of Health. The study funders and supporters had no role in study design, data collection and analysis, decision to publish, or preparation of the manuscript.

**Competing interests:** The authors have declared that no competing interests exist.

## Introduction

Sickle cell disease is among the most prevalent inherited hemoglobinopathies globally, with an estimated >300,000 affected births each year, and the largest burden is in sub-Saharan Africa [1–3]. In Uganda, the public health impact is substantial: the sickle cell trait prevalence has been estimated at 13.3%, and >20,000 children are born annually with SCD [4].

As survival improves through advances in newborn screening and wider access to disease-modifying therapy, particularly hydroxyurea, an increasing proportion of patients now survive into adolescence and young adulthood, shifting the clinical landscape from early childhood mortality to the long-term burden of chronic systemic morbidity [5,6].

Although the term "steady state" is commonly used in both research and clinical practice as a reference point for stable disease, it does not necessarily represent physiological normality. In this setting, patients may be free of overt acute complications yet continue to experience chronic hemolytic anemia, inflammation, endothelial dysfunction, and microvascular injury, processes central to cumulative organ injury in SCD [7]. However, even within such "stable" windows, chronic hematologic and biochemical alterations, anemia, leukocytosis, thrombocytosis, and hemolysis-related hepatic changes may persist and contribute to downstream systemic morbidity over time.

Adolescents and young adults represent a particularly critical subgroup because they are transitioning between pediatric and adult care, often with evolving adherence challenges and increasing cumulative exposure to SCD-related systemic injury. Despite this notion, the burden and spectrum of subclinical hemodynamic and hematological abnormalities are under-characterized in this population, especially in low- and middle-income countries such as Uganda, culminating in delayed detection and preventive care.

We therefore aimed to characterize the steady-state profile of adolescents and young adults with SCD attending the Mulago National Referral Hospital Sickle Cell Clinic, focusing on clinical parameters, anthropometry, laboratory indices, and treatment patterns.

## Materials and methods

### Ethical statement

Ethical approval was obtained from the Makerere University School of Biomedical Sciences Research and Ethics Committee (SBS-2023–479), with administrative approvals from Mulago National Referral Hospital (MNRH) and the Sickle Cell Clinic. Written informed consent was obtained from participants aged ≥18 years; for those aged 13–17 years, written assent was obtained alongside parent/guardian consent.

### Study design and setting

This was a hospital-based cross-sectional study conducted from 1 July to 30 September 2024 among adolescents and young adults with SCD attending the MNRH Sickle

Cell Clinic in Kampala, Uganda, with laboratory investigations performed at the Uganda Heart Institute laboratory, located within the Mulago Hospital Complex.

## Patient population

We recruited individuals aged 13–30 years living with SCD in Uganda, receiving care at MNRH Sickle Cell Clinic.

Participants were eligible if they had confirmed hemoglobin SS by hemoglobin electrophoresis, were attending routine outpatient follow-up, and were in steady state. Participants were excluded if they had documented congenital or acquired cardiac anomalies or had an ongoing sickle cell crisis at enrolment.

Participants were recruited using systematic random sampling: every third eligible clinic attendee was approached for enrolment until the target sample was reached.

## Definition of steady state

We defined steady state as no documented acute crisis for at least four consecutive weeks, and no blood or blood-product transfusion in the preceding three months at the time of enrolment.

## Sample size

The sample size was derived from the parent pilot study designed to assess the feasibility of clinical, laboratory, and cardiovascular assessments among adolescents and young adults with SCD. We estimated the sample size following the approach proposed by Viechtbauer et al. for pilot studies [8]. Using this method, and assuming a conservative event probability of 5% at a 95% confidence interval, the minimum required sample size was calculated as:

$$\frac{ln(1-\gamma)}{ln(1-\pi)} \tag{1}$$

where $n$ is the sample size, $\pi$ represents the assumed probability of observing the event of interest, and $\gamma$ denotes the confidence level. Substituting the assumed values yielded a minimum sample size of 59 participants. Accordingly, we enrolled 60 adolescents and young adults with SCD.

## Data collection procedures and measurements

We collected data using a coded, pretested, interviewer-guided questionnaire administered in English or Luganda at the Sickle Cell Clinic consultation room. Recorded variables included sociodemographic characteristics (age, sex), anthropometry (body mass index), clinical characteristics (resting blood pressure and heart rate; self-reported vaso-occlusive crises and transfusions), and medication use (hydroxyurea, folic acid, antibiotic prophylaxis, and antimalarial prophylaxis), including dose and duration where available.

Body mass index (BMI) was calculated as weight in kilograms divided by height in meters squared. Nutritional status was classified using World Health Organization criteria [9,10]: BMI-for-age Z-score < −2 standard deviations for adolescents aged 13–18 years, and BMI < 18.5 kg/m² for participants aged ≥19 years.

## Laboratory measurements

After completing the questionnaire, participants were escorted to the Uganda Heart Institute for phlebotomy. Ten milliliters of venous blood were collected in the morning. Complete blood count (CBC) samples were transported immediately for analysis, while samples for liver chemistry were centrifuged within one hour; serum was analyzed promptly, with results typically released within two hours. A complete blood count was analyzed on a Mindray BC-6200 analyzer, and liver chemistry was analyzed on a Roche Hitachi Cobas c 311.

## Statistical analysis

Data were coded and stored securely, entered into Microsoft™ Excel™, and exported to R (version 4.4.2) for analysis. Categorical variables were summarized as frequencies and percentages. Continuous variables were summarized as mean±standard deviation or median (IQR), depending on distribution.

## Results

The mean age was 16.5±3.3 years, and 34 (56.7%) were female. The mean body mass index was 18.4±2.8 kg/m². Based on World Health Organization age-appropriate criteria, 32 participants (53.3%) were underweight. The proportion of underweight was higher among adolescents (56.5%) compared with young adults (42.9%) (Table 1).

Mean systolic blood pressure was 107.9±15.5 mmHg, and mean diastolic blood pressure was 60.3±12.6 mmHg. Mean heart rate was 89.5±15.5 beats/min (Table 2).

Mean hemoglobin concentration was 9.1±2.2 g/dL. Other laboratory indices are summarized in (Table 3).

Prophylactic and disease-modifying medication use is summarized in Table 4. Fifty-two (86.7%) participants reported hydroxyurea therapy. All participants reported folic acid supplementation and antimalarial prophylaxis (sulfadoxine–pyrimethamine).

Among participants receiving hydroxyurea, the median weekly dose was 5,500 mg (interquartile range [IQR]: 3,500–7,000), with a median treatment duration of 36 months (IQR: 12–72) (Table 5). Folic acid supplementation was administered at a fixed dose of 5 mg/day, with a median duration of use of 10 years (IQR: 7–14). Antimalarial prophylaxis was prescribed at a median daily dose of 2 tablets (IQR: 2–3), with a median duration of use of 9 years (IQR: 5–13).

## Discussion

Our study describes the steady-state clinical, laboratory, and treatment profile of adolescents and young adults with SCD attending a tertiary sickle cell clinic in Uganda. Even in a clinically stable period, we observed several findings consistent

**Table 1. Sociodemographic and anthropometric characteristics of the study participants (N=60).**

| Characteristic | Value |
| --- | --- |
| Age, years (mean±SD) | 16.5±3.3 |
| Female sex, n (%) | 34 (56.7) |
| BMI, kg/m² (mean±SD) | 18.4±2.8 |
| Weight, kg (mean±SD) | 46.5±10.1 |
| Underweight (overall), n (%) | 32 (53.3) |
| Underweight among adolescents, n/N (%) | 26/46 (56.5) |
| Underweight among adults, n/N (%) | 6/14 (42.9) |

*BMI, body mass index; SD, standard deviation. Underweight is defined using World Health Organization criteria: BMI-for-age Z-score<−2 SD for adolescents aged 13–18 years and BMI<18.5 kg/m² for adults aged ≥19 years.*

**Table 2. Hemodynamic characteristics at steady state.**

| Characteristic | Value |
| --- | --- |
| Systolic blood pressure, mmHg (mean±SD) | 107.9±15.5 |
| Diastolic blood pressure, mmHg (mean±SD) | 60.3±12.6 |
| Pulse pressure, mmHg (mean±SD) | 47.5±13.9 |
| Heart rate, beats/min (mean±SD) | 89.5±15.5 |

**Table 3. Laboratory characteristics at steady state.**

| Laboratory parameter | Mean ± SD |
|---|---|
| Total white cell count, × 10⁹/L | 10.55 ± 4.28 |
| Neutrophil count, × 10⁹/L | 5.60 ± 2.59 |
| Lymphocyte count, × 10⁹/L | 3.85 ± 1.77 |
| Monocyte count, × 10⁹/L | 0.82 ± 0.40 |
| Eosinophil count, × 10⁹/L | 0.26 ± 0.24 |
| Basophil count, × 10⁹/L | 0.05 ± 0.04 |
| Platelet count, × 10⁹/L | 456.47 ± 179.91 |
| Red blood cell count, × 10¹²/L | 3.16 ± 0.89 |
| Hematocrit, % | 28.30 ± 6.67 |
| Hemoglobin, g/dL | 9.14 ± 2.23 |
| Mean corpuscular volume, fL | 91.17 ± 10.93 |
| Mean corpuscular hemoglobin, pg | 29.47 ± 3.84 |
| Mean corpuscular hemoglobin concentration, g/dL | 32.28 ± 0.92 |
| Red cell distribution width, % | 20.56 ± 4.61 |
| Total protein, g/dL | 8.38 ± 0.70 |
| Albumin, g/dL | 4.63 ± 0.41 |
| Aspartate aminotransferase, U/L | 42.54 ± 16.94 |
| Alanine aminotransferase, U/L | 20.31 ± 11.31 |
| Alkaline phosphatase, U/L | 194.22 ± 85.58 |
| Gamma-glutamyl transferase, U/L | 33.43 ± 25.62 |
| Direct bilirubin, mg/dL | 0.59 ± 0.29 |
| Total bilirubin, mg/dL | 2.87 ± 2.44 |

**Table 4. Prophylactic and disease-modifying medication use among study participants.**

| Characteristic | Value |
|---|---|
| Hydroxyurea, n (%) | 52 (86.7) |
| Folic acid, n (%) | 60 (100.0) |
| Antimalarial prophylaxis, n (%) | 60 (100.0) |

**Table 5. Dose and duration of prophylactic and disease-modifying medications among study participants.**

| Medication variable | Median (IQR) |
|---|---|
| Folic acid dose (mg/day) | 5 (5 –5) |
| Duration of folic acid use (years) | 10 (7 –14) |
| Fansidar daily dose (tablets) | 2 (2 –3) |
| Duration of sulfadoxine–pyrimethamine use (years) | 9 (5 –13) |
| Weekly hydroxyurea dose (mg) | 5,500 (3,500–7,000) |
| Duration of hydroxyurea therapy (months) | 36 (12–72) |

with the chronic SCD phenotype: low body mass index with a high proportion of participants who were underweight, a hemodynamic pattern characterized by relatively low blood pressure and higher heart rate, persistent anemia, leukocytosis and thrombocytosis, and biochemical evidence compatible with ongoing hemolysis. Treatment patterns reflected

contemporary clinical practice in Uganda, with high reported hydroxyurea use and near-universal supportive therapy [11]. These findings reinforce an important clinical point: "steady state" should be interpreted as the absence of acute events, not the absence of pathophysiological processes.

More than half of the participants in this study population were underweight, highlighting the nutritional vulnerability of adolescents and young adults with SCD in Uganda. Growth impairment and low BMI are well-described in several regions across Africa, where SCD-related hypermetabolism, chronic inflammation, recurrent infections, and micronutrient deficits intersect with household food insecurity and limited dietary diversity. A 2021 systematic review focusing on African data analyzed several studies, particularly from West Africa, showing that children and adolescents with SCD frequently have lower BMI and are often underweight compared with controls [12]. This pattern has also been reported in hospital-based studies in Uganda, where undernutrition among children with SCD is common and associated with clinical and social determinants [13,14].

Compared with data from high-income countries (HICs), underweight appears more prominent in Africa and other low- and middle-income countries (LMICs), likely reflecting the combined burden of SCD-related morbidity and contextual constraints on nutrition and supportive care. Such data also suggest that BMI distributions in SCD vary substantially across regions and care environments, supporting the need for locally grounded reference data rather than assuming HICs phenotypes generalize to African settings [15]. From a practice standpoint, the high prevalence of underweight in our study population argues for routine nutritional screening and targeted interventions as part of steady-state care (dietary counseling, micronutrient assessment where feasible, and structured follow-up), particularly during the transition from pediatric to adult services when adherence and continuity of care may decline.

Our study population demonstrated a hemodynamic profile consistent with results from prior literature about SCD hemodynamics: relatively lower systolic and diastolic blood pressures alongside higher heart rates. Evidence supports that patients with SCD, on average, have lower blood pressure than healthy controls, while pulse rate and pulse pressure may be higher, findings often attributed to chronic anemia-related high cardiac output, reduced systemic vascular resistance, and autonomic and vascular adaptations over time [16]. In Africa, similar observations have been reported in adults with SCD at steady-state, including data from a Nigerian study demonstrating lower arterial blood pressure indices in SCD compared with matched controls [17].

These hemodynamic characteristics are clinically relevant for two reasons. First, they can lead to under-recognition of "relative hypertension" in SCD patients; they may remain within conventionally "normal" ranges yet still experience adverse outcomes when blood pressure rises above their lower baseline. Second, a higher resting heart rate may reflect chronic anemia and physiologic stress, and in some settings has been linked to elevated cardiopulmonary risk markers. While our current study is descriptive and cannot infer prognosis, the steady-state hemodynamic profile supports integrating routine blood pressure and heart-rate surveillance into steady-state visits and interpreting values in the context of SCD-specific baselines rather than generic population norms.

The mean hemoglobin concentration in our study (~9 g/dL) aligns with steady-state anemia patterns described across Africa and in HICs. In LMICs, steady-state hemoglobin typically remains low because of chronic hemolysis and marrow stress, often compounded by infections, nutritional deficiencies, and variable access to comprehensive care. In HICs and in controlled trial settings, hydroxyurea therapy commonly increases hemoglobin into the ~9 g/dL range without significant cytopenias, especially with dose optimization and monitoring [18,19]. In sub-Saharan Africa, evidence from a randomized trial also demonstrates that hydroxyurea improves hematologic parameters and clinical outcomes in children, supporting its expanding role as standard-of-care as access improves [6]. Given the high hydroxyurea uptake in our study population, the observed hemoglobin level is biologically plausible and may reflect, at least in part, disease-modifying therapy in a real-world setting; however, causal attribution is not possible in this cross-sectional analysis.

Leukocytosis and thrombocytosis were also prominent, consistent with "steady-state inflammation" and functional asplenia in SCD. Studies from Africa describing steady-state hematology similarly report elevated white cell and platelet

counts among patients with SCD compared with healthy controls [20]. Importantly, data from HICs indicate these elevations are not merely laboratory curiosities. Curtis et al.'s study in adults showed that higher steady-state leukocyte and platelet counts were associated with frequent emergency care utilization, underscoring that persistent inflammation and platelet activation may track with disease severity and health-care burden [21]. In LMIC contexts, where baseline infection pressure is higher and access to preventive care may be inconsistent, steady-state leukocytosis may be amplified by recurrent or subclinical infections, malaria exposure, and chronic inflammatory stimulation. Clinically, these findings support routine CBC monitoring even in steady state, with interpretation that distinguishes chronic SCD patterns from concurrent infection, and careful review of adherence to disease-modifying therapy and supportive interventions.

The liver function panel profile in our study population is consistent with ongoing hemolysis and chronic hepatobiliary stress, common even outside acute crises. In SCD, hemolysis contributes to elevated bilirubin and other hemolytic markers, and chronic hemolysis-related pathways have been well characterized in hemolysis indices used in observational research [22]. Contemporary mechanistic and clinical studies also support the expected correlations between hemolysis-related biomarkers such as AST and total bilirubin, reinforcing that "steady state" remains a hemolytic state [23].

Data from LMICs further suggest that elevated bilirubin is frequently reported in steady state, reflecting sustained red-cell turnover and pigment load, often alongside increased risks of cholelithiasis and other hepatobiliary complications over time [24]. While our study did not include imaging or detailed hepatology phenotyping, the biochemical pattern supports incorporating periodic liver chemistry monitoring into steady-state care, particularly in patients with symptoms suggestive of hepatobiliary disease, prior transfusion exposure, or suspected viral hepatitis. Interpretation should remain cautious, as elevated transaminase levels can reflect hemolysis, hepatic congestion, infection, medication exposure, or coexisting liver disease, and cross-sectional testing cannot distinguish these etiologies without longitudinal trends and clinical correlation.

Hydroxyurea use was reported by 86.7% of participants, higher than many historical reports from LMICs and consistent with the evolving Ugandan context in which hydroxyurea has become more integrated into routine care [25–27]. Qualitative and mixed-methods research from Uganda highlights that hydroxyurea uptake has increased but remains influenced by patient-level barriers (knowledge gaps, perceived toxicity, monitoring concerns, cost, and access) [28]. Our estimate likely reflects the specific characteristics of a tertiary clinic population and the study's focus on adolescents and young adults in steady state (patients engaged in longitudinal care, potentially more likely to be on hydroxyurea). Differences in uptake across Africa are expected, given variation in national policy, drug supply chains, laboratory monitoring capacity, and provider experience.

Folic acid use was universal in our study population, reflecting common supportive practice in SCD programs. In contrast, no participant reported antibiotic prophylaxis. This is expected given the study population's age distribution: major guidelines recommend penicillin prophylaxis primarily in early childhood, generally discontinued around age 5 when immunizations are up to date, although practice varies [29,30]. The absence of prophylactic antibiotics in an adolescent and young adult population is therefore plausible and should not be interpreted as a care gap without an age-specific context.

Antimalarial prophylaxis was also universally reported. In malaria-endemic regions, malaria prevention is a cornerstone of SCD care, given the high morbidity associated with malaria infection in SCD. Uganda Clinical Guidelines include malaria prophylaxis strategies for SCD, including sulfadoxine–pyrimethamine schedules [11]. In our study population, recorded dosing should be interpreted cautiously because antimalarial prophylaxis is often administered on monthly schedules rather than as a daily dose; nonetheless, the key message is that malaria prevention was embedded within routine care for this study population.

Overall, the phenotype observed here is directionally consistent across settings, anemia, leukocytosis, thrombocytosis, and hemolysis persist in steady state globally, but the magnitude and clinical context differ. In HICs, comprehensive care (early diagnosis, vaccination, penicillin in childhood, routine monitoring, and broader access to hydroxyurea and transfusion support) can attenuate infection burden and may modify hematologic and growth trajectories [19,30]. In several LMICs, environmental exposures (notably malaria), nutritional constraints, and variable diagnostic capacity may

accentuate underweight and inflammatory markers while delaying detection of subclinical complications [12]. The high hydroxyurea uptake in this study population in Uganda suggests that when hydroxyurea becomes routinely available and acceptable to patients, steady-state hematologic profiles may begin to resemble those reported in optimized care settings, though broader generalizability requires multi-site data.

Our study has several strengths and limitations. A key strength of our study is its focus on adolescents and young adults, a transition-age group that remains under-described in many African literature, using standardized hospital-based measurements and a pragmatic steady-state definition aligned with routine care. However, several limitations should guide interpretation. First, the cross-sectional design precludes causal inference (e.g., hydroxyurea cannot be credited for specific hematologic profiles). Second, this is a single-site tertiary hospital study population and may not represent patients receiving care at lower-level facilities or those disengaged from care. Third, some treatment variables rely on self-report and clinic documentation, which can introduce misclassification (particularly for dosing schedules). Finally, we did not include a healthy control comparison group, which limits the inference about the degree of deviation from local population norms.

Despite these limitations, the findings have practical implications. The high prevalence of underweight individuals supports the integration of nutrition assessment and support into steady-state visits. The hemodynamic profile reinforces the need for careful longitudinal tracking of blood pressure and heart rate, interpreted against SCD-specific baselines. Leukocytosis and thrombocytosis suggest the need for routine CBC monitoring and interpretation to distinguish chronic inflammation associated with SCD from concurrent infections. Liver function patterns consistent with hemolysis support periodic biochemical monitoring and clinical evaluation for hepatobiliary complications. Finally, the observed treatment patterns underscore the feasibility of implementing disease-modifying therapy and structured supportive care in a Ugandan tertiary care setting, while also highlighting the need for implementation research on sustained access, adherence, and monitoring across the health system.

## Conclusion

Adolescents and young adults with SCD at steady state exhibit persistent hematologic and biochemical abnormalities, altered hemodynamics, and a high proportion of undernutrition. High hydroxyurea uptake reflects expanding access to disease-modifying therapy. These findings not only provide a baseline reference but also support integrated steady-state care that combines routine monitoring, nutritional assessment, sustained treatment delivery, and future research in Uganda.

## Acknowledgments

We are grateful to Norah Kedde for assistance in recruiting the study participants.

## Author contributions

**Conceptualization:** Abdul Magala Ssekandi, Haruna Muwonge, Robert Kalyesubula.

**Data curation:** Abdul Magala Ssekandi, Ian Guyton Munabi.

**Formal analysis:** Abdul Magala Ssekandi, Haruna Muwonge, Agnes Namaganda, Ivan Kimuli, Roselyne Akugizibwe, Hassan Kasujja, David Mukunya, Grace Ndeezi, Sarah Kiguli.

**Funding acquisition:** Abdul Magala Ssekandi, Ruth Namazzi, Robert Kalyesubula, Ian Guyton Munabi, Christine Sekaggya-Wiltshere, David Mukunya, Grace Ndeezi, Sarah Kiguli.

**Investigation:** Abdul Magala Ssekandi, Ruth Namazzi, Haruna Muwonge, Robert Kalyesubula, Roselyne Akugizibwe, David Mukunya, Grace Ndeezi.

**Methodology:** Abdul Magala Ssekandi, Ruth Namazzi, Haruna Muwonge, Robert Kalyesubula, Christine Sekaggya-Wiltshere, Agnes Namaganda, Ivan Kimuli, Roselyne Akugizibwe, Hassan Kasujja, David Mukunya, Grace Ndeezi, Sarah Kiguli.

**Project administration:** Abdul Magala Ssekandi, Haruna Muwonge, Ian Guyton Munabi, Agnes Namaganda, Ivan Kimuli, Grace Ndeezi, Sarah Kiguli.

**Resources:** Abdul Magala Ssekandi, Ruth Namazzi, Robert Kalyesubula, Christine Sekaggya-Wiltshere, Ivan Kimuli, Grace Ndeezi, Sarah Kiguli.

**Software:** Abdul Magala Ssekandi.

**Supervision:** Ruth Namazzi, Haruna Muwonge, Robert Kalyesubula, Ian Guyton Munabi, Christine Sekaggya-Wiltshere, Ivan Kimuli, David Mukunya, Grace Ndeezi, Sarah Kiguli.

**Validation:** Sarah Kiguli.

**Visualization:** Robert Kalyesubula, Hassan Kasujja.

**Writing – original draft:** Abdul Magala Ssekandi, Agnes Namaganda, Roselyne Akugizibwe, Hassan Kasujja.

**Writing – review & editing:** Abdul Magala Ssekandi, Ruth Namazzi, Haruna Muwonge, Robert Kalyesubula, Ian Guyton Munabi, Christine Sekaggya-Wiltshere, Agnes Namaganda, Ivan Kimuli, Roselyne Akugizibwe, Hassan Kasujja, David Mukunya, Grace Ndeezi, Sarah Kiguli.

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
