## [Decision Letter · Decision Letter 0]

17 Feb 2026

PGPH-D-26-00114

Clinical and Laboratory Characteristics of Adolescents and Young Adults with Sickle Cell Disease at Steady State in Uganda

Dear Dr. Ssekandi,

Thank you for submitting your manuscript to PLOS Global Public Health. After careful consideration, we feel that it has merit but does not fully meet PLOS Global Public Health’s publication criteria as it currently stands. Therefore, we invite you to submit a revised version of the manuscript that addresses the points raised during the review process.

The manuscript is well written and addresses an important issue of abnormal parameters even when the individuals with sickle cell disease are expected to be in steady state. The study focuses on adolescents and young adults, a group not well studied normally. Both reviewers have recommended minor revision. The authors are requested to address the specific suggestions given by the reviewers including commenting on the aspects involving nutrition & disease modifying therapy in the discussion section. Once these changes are incorporated in the revised manuscript, it may be accepted.

We look forward to receiving your revised manuscript.

Kind regards,

Debjani Paul

Academic Editor

Journal Requirements:

Additional Editor Comments (if provided):

Both reviewers have recommended minor revision. The authors are requested to address the specific suggestions given by the reviewers including commenting on the aspects involving nutrition & disease modifying therapy in the discussion section.

Reviewers' comments:

Reviewer's Responses to Questions

**Comments to the Author**

1. Does this manuscript meet PLOS Global Public Health’s publication criteria ? Is the manuscript technically sound, and do the data support the conclusions? The manuscript must describe methodologically and ethically rigorous research with conclusions that are appropriately drawn based on the data presented.

Reviewer #1: Yes

Reviewer #2: Yes

2. Has the statistical analysis been performed appropriately and rigorously?

Reviewer #1: Yes

Reviewer #2: Yes

3. Have the authors made all data underlying the findings in their manuscript fully available (please refer to the Data Availability Statement at the start of the manuscript PDF file)?

Reviewer #1: Yes

Reviewer #2: Yes

4. Is the manuscript presented in an intelligible fashion and written in standard English?

Reviewer #1: Yes

Reviewer #2: Yes

Reviewer #1: On review of the literature available this study appears to present the results of original research on this subject. In addition, it appears to be a valuable continuation of work at the Mulago Hospital to improve patient outcomes for adolescents and young adults with sickle cell disorder.

While a couple of preprint servers have this study, there is no evidence of publication elsewhere that the reviewer could locate in the time available for this review.

Data analysis is straight forward and uncomplicated.

The conclusion is brief and clear in its recommendation but misses out some valid findings in the discussion relating to nutrition & disease modifying therapy thus reducing patient evaluation to a numerical exercise.

Please review the sentence in lines 39-40.

Ethical considerations have been documented in this study.

Reviewer #2: This is a well-written manuscript describing baseline "steady-state" laboratory data in a cohort adolescents and young adults living with sickle cell disease from a tertiary medical center in Uganda. Results are appropriately described in text and Tables. The discussion provides local context based on practice patterns, and comparisons to similar studies in both HIC and LMICs. This information should be of interest to the journal's readers and researchers. My questions to the authors are as follows:

1. Please correct the second sentence in the abstract.

2. Please provide a reference or describe the reference data used for determination of the % of underweight study participants

3. Please include mean and standard deviation for the weights of study participants in Table 1. This data would be helpful for assessing the hydroxyurea dosages, which are typically provided in mg/kg.

**Do you want your identity to be public for this peer review?** For information about this choice, including consent withdrawal, please see our Privacy Policy .

Reviewer #1: No

Reviewer #2: **Yes:** Carlton Dampier MDCarlton Dampier MD

---

## [Editor Report · Decision Letter 1]

13 Mar 2026

Clinical and Laboratory Characteristics of Adolescents and Young Adults with Sickle Cell Disease at Steady State in Uganda

PGPH-D-26-00114R1

Dear Dr. Ssekandi,

We are pleased to inform you that your manuscript 'Clinical and Laboratory Characteristics of Adolescents and Young Adults with Sickle Cell Disease at Steady State in Uganda' has been provisionally accepted for publication in PLOS Global Public Health.

Best regards,

Debjani Paul

Academic Editor

The authors have addressed all comments in the revised manuscript, which can now be accepted.